# Analysis of technical efficiency of smallholder tomato producers in Asaita district, Afar National Regional State, Ethiopia

**Dagmawe Menelek Asfaw**⬤ *

Department of Economics, College of Business and Economics, Samara University, Samara, Ethiopia

* dagmawemenelek@gmail.com

**Data Availability Statement:** All relevant data are within the manuscript and its Supporting Information files.

## Abstract

The tomato had nutritional, economic and health benefits to the societies, however, its production and productivity were low in developing countries and particularly in Ethiopia. This might be due to technical inefficiency caused by institutional, governmental, and farmers related factors. Therefore this study tried to investigate the factors that affecting technical efficiency and estimating the mean level of technical efficiency of tomato producers in Asaita district, Afar Regional State, Ethiopia. Both primary and secondary data sources were used; the primary data was collected from 267 tomato producers from the study area cross-sectional by using a multistage sampling technique. The single-stage stochastic frontier model and Cobb Douglas production function were applied and statistical significance was declared at 0.05. The maximum likelihood estimates of the stochastic frontier model showed that land, labor, tomato seed, and oxen have a significant effect on tomato output; and education, extension contact, training, and access to credit have a positive and significant effect on technical efficiency, whereas household size, off-farm income, livestock ownership, distance to market, and pesticides have a worthy and significant effect on technical efficiency; and also estimated mean technical efficiency of tomato producer in a study area was 80.9%. In a line with this, the responsible body should prioritize rural infrastructure development in areas such as education, marketplace, and farmer training centers; demonstrate access to credit and extension services; use the recommended amount of pesticides per hectare, and give more intension to mixed farming rather than animal husbandry exclusively.

## Introduction

Vegetables occupy approximately 1.64% (238,564 hectares) of total crop production land at the national level and account for approximately 2.08% (8.76 million quintals) of total crop production per season. Such production is primarily produced by small-scale producers (They are on average holding 0.11 hectares of land per person [2]) with an estimated 5.7 million farmers, and it is also produced by large-scale producers with fruit and vegetable processing carried out solely by state organizations [1,2].

**Funding:** The authors received no specific funding for this work.

**Competing interests:** The authors have declared that no competing interests exist

In Ethiopia's agriculture system the production of tomatoes (Solanum Lycopersicum Mill) was introduced dates back to the period between 1935 and 1940 [3,4]. Tomato was cultivated for commercial purposes at the upper Awash by Merti Agroindustry for both domestic as well as export markets in 1980 with a production area of 80 hectares for the first time [4].

However, the total cultivated area increased to 833 hectares in the year 1993, and later on, the cultivation spread towards other parts of the country [5]. Until 2016, the Ethiopian National Agricultural Research System (NARS) had developed approximately 39 tomato varieties [6]. Melkashola, Marglobe, Melkasalsa, Heinz 1350, Fetan, Bishola, Eshet, and Metadel are among the open-pollinated tomato varieties released by the Melkassa Agricultural Research Center (MARC) and are recommended for commercial and small-scale production in Ethiopia [7].

Tomato is being cultivated widely and planted in Ethiopia with about 700 to over 1400 mm annual rainfall, indifferent place, seasons, soils, weather conditions, technology (rain feed or irrigation), and output level [8,9], which is ranking fourth (0.35 million quantal) in terms of annual total national vegetable production after Ethiopian cabbage, red pepper, and green pepper are third in area coverage and 2.5% of land allotted to it from the vegetable production land [1,10,11]. In Ethiopia, tomatoes served as an ingredient in many dishes (local sauce), fresh produce is sliced and used as a salad; and processed products such as tomato paste, tomato juice, soups, stews, and tomato catch-up are consumed in large quantities when compared to other vegetables [12,13].

Tomato is the most vital vegetable in Ethiopia and it is the most important source of a healthy diet and home to micronutrients like- vitamin C, biotin, molybdenum, vitamin K, vitamin A (in the form of beta-carotene), vitamin B, vitamin E, folate, niacin [14–16]. It is also a good source of protein, chromium, pantothenic acid, molybdenum, choline, zinc, and iron [17].

Tomato is a high cash crop production and it has to be the source of income for small-scale farmers and provides employment opportunities in the production, distribution, and processing in the industries [4]. Tomatoes play an important role in Ethiopia's poverty and food insecurity programs because they have a short harvesting season, relatively high production per hectare (example 9.4 tons per hectare in 2016), and one of the strategic commodities prioritized by the Ethiopian government for agro-industry development [18]. It also strengthens the national economy as a source of raw materials for value-added agro-processing industries and foreign currency for an exportable tomato to international markets (For example, $9.006 million revenue from both fresh and chilled tomato export in 2017) [19,20]. Therefore such and other issues make the tomato is the most vital vegetable in Ethiopia.

Despite its nutritional, economic, and health importance; and has large potential to produced tomato, its production and productivity were low in Ethiopia as a general [20,21], for instance, in 2016 the total cultivated area was approximately 9700 hectares, total production of 91300 tons of fresh tomato, and average productivity of 9.4 tons per hectare [18]. In general, the average productivity of tomatoes in Ethiopia is too low (for instance, 6.18 tons/hectare in 2018 and 6 tons/hectare in 2015) compared with the world average productivity of 38.3 tons/ hectare and also 16, 96.8, 63.9, 43 and 38.3 tons/hectare in Africa, America, Europe, Asia in 2018, respectively [4,22,23].

The major responsible factors for low yield, productivity, and inconsistent production were: shortage of improved seed, pesticides [24], fertilizer [25,26], unreliable rainfall, biotic and abiotic factors [27], price fluctuation after harvest, product nature(perishability) [19], post-harvest loss [17]. This poor production and productivity of tomato outputs resulted in food insecurity [28], in turn, such problems have their own effect on the aggregated macroeconomic as a whole. One of the best mechanisms to mitigate food insecurity and improved

macroeconomic performance could be increasing productivity through enhancing production efficiency by addressing factors that hindering/improving efficiency [29]. Many scholars had been estimated the level of technical efficiency and investigated determinants of the technical efficiency of vegetable products in general and tomatoes in particular. For example, a study conducted by [11,13,20,30–32] revealed that the technical efficiency of vegetables (tomatoes) can be determined by: farm management, infrastructural development, transportation access, extension contact, access of training, irrigation accessibility, membership in a farmers' association, and credit availability.

Households with higher educational levels and household size had relatively more technical efficiency [33–36], whereas households with off-farm income and ownership of livestock were less technically inefficient compared to their counterparts [36–38].

However most of such studies were used a two-stage stochastic frontiers model approach rather than a single-stage approach to estimate the level of technical efficiency, single staged approach is to preserve consistency [39], it guarantees that the distributional assumption of inefficiency error term [40] and socio-economic variables (determinants of technical efficiency) may have a direct influence on the production efficiency [41]. In addition, the result of such studies was exclusive and inconsistent with the study area Asaita district, This is because, in the study area (Asaita district) the smallholder tomato producers were different from previously studied area smallholder tomato producers, due to, they were used different technology (i.e. fully irrigated), operate their production at different agro-ecological zone (i.e. tropical), different know-how of production and societal way of life(i.e. agro-pastoralist). Therefore, this study used a single-stage approach to estimate technical efficiency and analysis the technical efficiency of small scale tomato producers in Asaita district.

## Materials and methods

### Description of the study area

The study was conducted at Tendaho Irrigation Project (TIP) in Asaita district, which is situated in the Lower Awash Valley of the Afar National Regional State (ANRS), northeastern Ethiopia. The Project is located about 600 km away from Addis Ababa. It is situated at 11˚ 40' 77"N and 40˚57'49"E between Dubti and Asaita Districts. Asaita district is one of the districts in the Afar region of Ethiopia. According to Central Statistical Agency (CSA), [42], Asaita is part of the administration of Awisi zone. In this district there are 13 kebelles(It is the smallest administrative unit in Ethiopia) of those 11 are rural and the remaining 2 are urban kebelles [43]. The district has a latitude and longitude of 11˚34′N 41˚26′E and an elevation of 300 meter (980 ft). In the district, a pastoral and agro-pastoral system of livestock production is the dominant practice. The mean temperature is between 30˚C and 45˚C per annum [44].Sampling technique and sample size.

The sampling technique in this study was the multistage sampling technique. In the first stage, from a total of 14 irrigation districts in Afar regional state, Asaita irrigation district was purposively selected due to the total production of tomatoes per production season and long-year experience in tomato production. In the second stage, From the total 13 kebeles found in Asaita district, only 5 kebeles were selected purposefully (because only those 5 kebeles were produced tomato). In the third stage, 267 households were selected using systematic random sampling from those 5 kebeles tomato producers population obtained from Asaita district agricultural office. The sample allocation to each kebelles was based on probabilities proportional to the total sampled population on the 2019/2020 production season (see Table 1). The intended sample sizes (267) were determined by using Kothari [45] sample size determination

**Table 1. Tomato growing farmers and sample size.**

| Tomato Producer Kebeles | Total no of Tomato Producers | Sampled Tomato Producers | |
|---|---|---|---|
| | | Number | Percentage |
| Berga | 149 | 52 | 19.6 |
| Hinle | 146 | 52 | 19.3 |
| Kerbuda | 141 | 49 | 18.5 |
| Kerdura | 161 | 57 | 21.2 |
| Mamule | 163 | 57 | 21.4 |
| Total | 760 | 267 | 100 |

Source: Asaita District Agricultural Office and own computation, (2019/2020).

formula.

$$\text{n} = \frac{Z^2 pq}{e^2} = \frac{(1.96)^2 (0.5)(0.5)}{(0.06)^2} = 266.7 \approx 267$$

Where: n = the sample size; Z = confidence level (Z = 1.96); $p = 0.5$; $q = 1 - p$; e = margin of error (0.06).

## Data sources and methods of data collection

Both primary and secondary data were gathered to meet the objective. Primary data was principally used, which was collected from a sample representative of tomato producers from Asaita irrigation site by questionnaire and semi-structured interview through a team of five trained enumerators for each sampled kebelles. Secondary data sources are also governmental and non-governmental institutions including both published and unpublished documents like rural development agricultural office, Ministry of Agriculture and Rural Development, and CSA, and other relevant information sources were used.

## Method of data analysis and model specifications

The tools for analyzing the data in this study were both descriptive and inferential (econometrics model) statistical analysis. Descriptive statistical analysis was used to analyze the survey data using measures of dispersion and central tendency like:—percentage, frequency, minimum, maximum, mean, and standard deviation. Regarding econometrics analysis, a single-stage stochastic frontier model was used to estimate the level of efficiency and analysis the determinants of technical efficiency of small-scale tomato producers.

The reason to choose a stochastic frontier model was, it is suitable for analyzing farm-level data where measurement errors are substantial and the weather(natural hazards, unexpected weather conditions, pest, and disease) is likely to have a significant effect [28,40] and it is also referred to as the econometric frontier approach, specifies a functional form for the cost, profit, or production relationship among inputs, outputs, and environmental factors, and it allows for random errors [46]. The Stochastic Frontier Approach (SFA) was developed independently by Aigner et al. [47] and Meeusen and Van der Broeck [48]. SFA approach can be extended to measure inefficiencies in individual production units based on some distributional assumptions for the technical and economic inefficiency scores.

Another issue regarding the stochastic frontier model is choosing the best fitted functional forms (Cobb-Douglas, constant elasticity of substitution, Translog, and other functional forms). By doing so, this study chose the Cobb–Douglas functional forms as the most fitted

functional form, with the help of the generalized likelihood-ratio test (LR). A Cobb–Douglas functional form is computational feasibility, simple, it is also convenient in interpreting elasticity of production, and it is very parsimonious concerning degrees of freedom [49–51].

The stochastic frontier production function can be express as Eq (1):

$$Y_i = F(X_i\beta)\exp(V_i - U_i) \qquad i = 1, 2, 3, - - -, 267 \tag{1}$$

Where:- $Y_i$ = Tomato output, $i$ = the $i^{th}$ farmer in the sample, $X_i$ = a vector of inputs used by the $i^{th}$ farmer, $\beta$ = a vector of unknown parameters, $V_i$ = a random variable which is assumed to be normally and independently distribute and $U_i$ = farm-specific technical inefficiency in production and nonnegative random variable.

The Cobb–Douglas form of stochastic frontier production is stated as Eq (2):

$$lnY = \beta_0 + \sum_{j=1}^{5}\beta_j lnX_{ij} + V_i - U_i \tag{2}$$

Where:- $ln$ = natural logarithm, $X_{ij}$ = is the quantity of input j used in the production process including oxen, labor, land, and quantity of seed and tractor.

After estimating the technical inefficiency ($U_i$) from Eq (2), the technical inefficiency model was specified as Eq (3):

$$\begin{aligned} U_i = \alpha_0 &+ \alpha_1(Edu_i) + \alpha_2(HHsize_i) + \alpha_3(Extc_i) + \alpha_4(Offincome_i) + \alpha_5(Trning_i) + \alpha_6(Credit_i) \\ &+ \alpha_7(Ownlvs_i) + \alpha_8(DSTmkt_i) + \alpha_9(Memcoop_i) + \alpha_{10}(DSTplot_i) + \alpha_{11}(Pest_i) \\ &+ \alpha_{12}(Expr_i) + \alpha_{13}(Sex_i) + \varepsilon_i \end{aligned} \tag{3}$$

**Where**:- $Edu_i$ Stands for the education level of households and it is a continuous variable referring to the years of schooling of the household head; $HHsize_i$ represents the number of family sizes in a given household; $Extc_i$ refers frequency of extension service by extensions agents during the production season; $Offincome_i$ represents Off/non-farm activities, it is a dummy variable (1 if the household was involved in off/non-farm activities and zero otherwise); $Trning_i$ represents the training of household heads related to potato production, 1 if the farmer gets training on tomato production, 0 otherwise; $Credit_i$ represent access to credit it is dummy variable indicating 1 if a farmer received and used credit in red pepper production and zero otherwise; $Ownlvs_i$ represent ownership of livestock and measured by Tropical Livestock Unit (TLU); $DSTmkt_i$ represents the distance of the nearest market in kilometers; $Memcoop_i$ represent the membership of farmer cooperative, it is a dummy variable (1 if the farmers were member, 0 otherwise); $DSTplot_i$ represent the distance from farmer home to tomato plot in minutes; $Pest_i$ represent usage pesticides in a litter for tomato production, it is a dummy variable (1 if the farmers used pesticide,0 otherwise); $Expr_i$ represent farmer experience in the production of tomato it is a continuous variable and measured in years; $Sex_i$ represent the sex of the household head, it is a dummy variable (0 if household head is female and 1 otherwise).

Farm specific technical efficiency was specified as Eq (4):

$$TE_i = \frac{Y}{Y^*} = \frac{Y}{f(X_i\beta)} \tag{4}$$

Where: $Y$ = Actual tomato output, $Y^*$ = potential tomato output and $TE_i$ = farm specific technical efficiency.

Variance parameters obtained from maximum likelihood estimation are expressed in Eqs (5) and (6):

$$\sigma^2 = \sigma^2{}_V + \sigma^2{}_U \tag{5}$$

$$\gamma = \frac{\sigma^2{}_U}{\sigma^2} \tag{6}$$

Where:- $\sigma^2$ is the total variance of the model, $\sigma^2{}_V$ = variance of the random error term, $\sigma^2{}_U$ = variance of the inefficiency error term and $\gamma$ = the term represents the ratio of the variance of inefficiency's error term to the total variance of the two error terms.

To analyze the determinant of technical efficiency, a single-stage estimation procedure was applied from a stochastic frontier production function using the maximum likelihood procedure. In single-stage estimation, inefficiency effects are defined as an explicit function of certain factors specific to the firm, and all the parameters are estimated in one step using the maximum likelihood procedure. One staged approach is to preserve consistency [39], it guarantees that the distributional assumption of inefficiency error term [40] and socio-economic variables (determinants of technical efficiency) may have a direct influence on the production efficiency [41].

**Ethical consideration and consent to participate.** Ethical clearance was obtained from the College of Business And Economics, Samara University. Confidentiality of the information was secured by excluding respondents' identifiers, such as names, from the data collection format. Finally, verbal informed consent was obtained from those who were smallholder tomato producers in Asaita district and willing to participate in the study. Moreover, the results were recommended to be disseminated by the responsible bodies who were involved in agriculture, agricultural extension services, agricultural NGOs.

## Results and discussion

### Descriptive analysis

The average household members that live in one house for the sample households were about 7.27 persons that ranging between 2 and 12 persons. The average age of the sample household heads was 39.88 years with a maximum of 70 and a minimum of 20 years old. This showed that the mean ages of the sampled households were within the range of economically active age and they were more energetic (Table 2).

The average farming experience and education level of the sample farmers in tomato production was 33.22 years and 4.06 grades ranging from 0 to grade 10, respectively, this indicated that they had a long farming experience and very low educational level (Table 2).

From the total sample household, 70.75 percent of them were male-headed households, whereas the remaining also female-headed households. Based on their marital status more

**Table 2. Demographic characteristics of sample households.**

| Variables | Mean | Standard Deviation | Minimum | Maximum |
|---|---|---|---|---|
| **Household size** | **7.27** | **7.99** | **2** | **12** |
| Age | 39.88 | 11.85 | 20 | 70 |
| Farming Experience | 33.22 | 15.03 | 9 | 52 |
| Education level | 4.06 | 3.02 | 0 | 10 |

Source: Own survey (2021).

than half of the samples were engaged in married (66.04 percent), this was the reason that the average household size was more than seven and it was also helpful for the family labor force for tomato production. The remaining sampled households were single and divorced, which were accounted for 23.58 and 10.38 percent of total sampled households, respectively (see Fig 1).

Fig 1Based on Table 3 the sample households were owned 2 hectares of land on average with a maximum of 3 hectares and a minimum of 0.23 hectares, this was showed that they were small-scale farmers and should be used resources(land) in a technically efficient manner. Other factors of production included labor, oxen, and tractors, which were used on average 20 labor days per hectare, 7.69 oxen days per hectare, and 2.95 tractor days per hectare. This

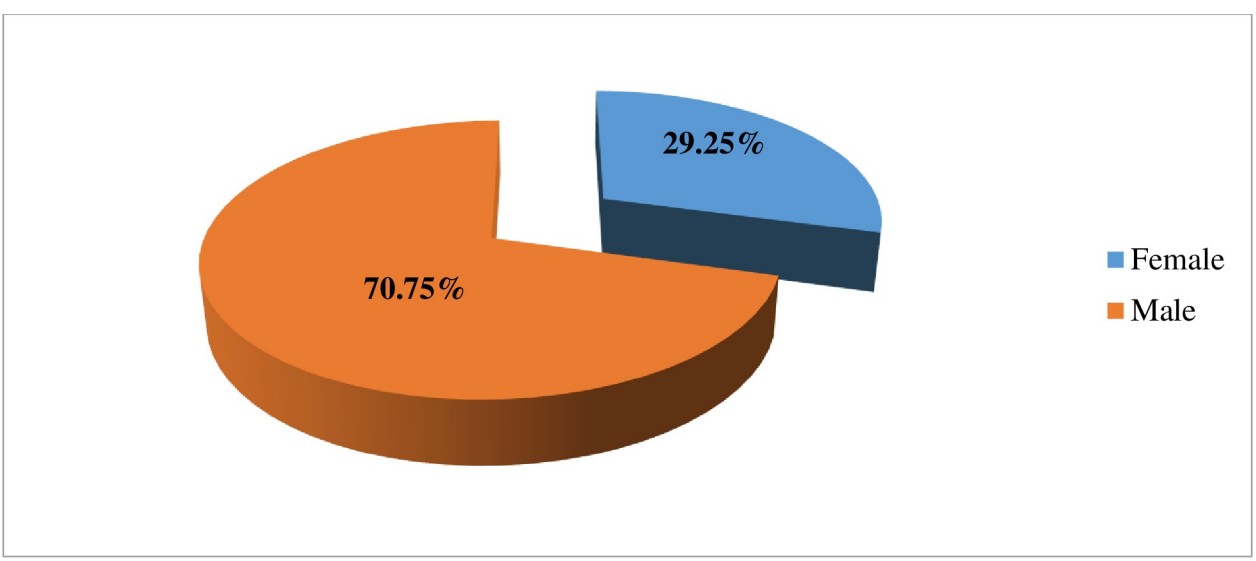

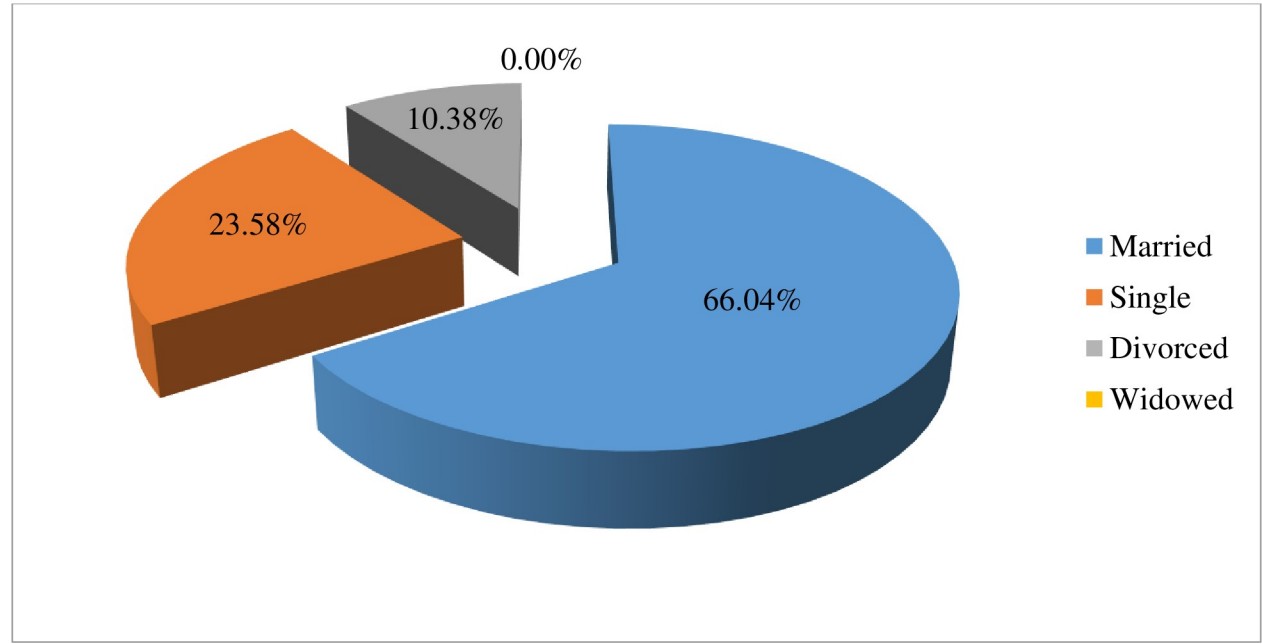

**Fig 1. Demographic characteristics of sample households.**

**Table 3. Stochastic frontier production function variables summery.**

| Variables | Observation | Mean | Standard Deviation | Minimum | Maximum |
|---|---|---|---|---|---|
| Output(Kg/ha) | 267 | 5,987.23 | 5,090.25 | 4,626 | 8,753.6 |
| Seed (kg/ha) | 267 | 1.02 | 0.23 | 0.83 | 4.35 |
| labor (man-day/ha) | 267 | 20 | 14.24 | 12 | 55 |
| Oxen (oxen-day/ha) | 267 | 7.69 | 6.85 | 6.54 | 8.36 |
| Tractor(tractor-day/ha) | 267 | 2.95 | 2.49 | 2.33 | 3.70 |
| Land (ha) | 267 | 2 | 0.53 | 0.5 | 3 |

Source: Own survey (2021).

justified that tomato production techniques were labor-intensive and should use capital more technically efficient way and also there was high variability in the usage of such inputs per hectare among sampled households. This also told us, some farmers were technically inefficient during tomato production than other farmers.

All sample households apply improved tomato seed (Roma VF) with an average of nearly 1.02 kilogram per hectare and the range was from a minimum of 0.83 kilograms per hectare to a maximum of 4.35 kilogram per hectare (Table 3). This highlighted mismanagement, technical inefficiency, and cost-ineffective usage of tomato improved seed among the sampled households.

Finally, Table 3, told us there were rooms of technical inefficiency in the production of tomato in sample housed, why because some households were produced 4,626 kilograms of tomato for a unit hectare of land and other households also produced 8,753.6 kilograms of tomato for a unit hectare, this implies that some households were produced below potential output level (8,753.6 kg/ha).

## Inferential analysis

**Pre-estimation test.** To achieve the objective of this study, the stochastic production frontier model was applied using the maximum likelihood estimation techniques with STATA 14 statistical software. Nonetheless, before proceeding to estimation and analysis, there were pre-estimation tests to ensure that the data fit with the stochastic production frontier model.

First, check whether there were problems of multicollinearity or not among continuous and categorical explanatory variables using the variance inflation factor (VIF) and contingency coefficient (CC), respectively. Both correlation (variance inflation factor) and association (contingency coefficient) results of the model were less than 10 and 0.75, therefore there was not a problem of multicollinearity in the model [52].

Second, choosing between traditional average production function (OLS) best fits the data set and the stochastic frontier model (SFM) by using a log-likelihood ratio test. In such a way the null hypothesis stated that $H_0{:}\gamma = \alpha_0,\alpha_1,---\alpha_9 = 0$ and the alternative hypothesis also $H_1$: $\gamma = \alpha_0,\alpha_1,---\alpha_9 \neq 0$, decision: if log-likelihood ratio test is greater than the $Chi-square(x^2)$ tabulated value at one degree of freedom and 5% significance level, reject the null hypothesis. The log-likelihood ratio test statistic is calculated as $LR = -2[lnL(H_0)-L(H_1)] = -2[-247.56 -(-214.3)] = 66.52$, this statistical value is higher than $Chi-square(x^2)$ tabulated value at one degree of freedom and 5% significance level, i.e. 3.84 (Table 4). Therefore, the null hypothesis is rejected and the stochastic frontier model is the best model to fit the data compared to the average production function method (OLS).

Third, select the appropriate functional form to fit the data of tomato using likelihood ratio test, in most case researchers were used Cobb-Douglas and Trans-log production function.

**Table 4. Summary of hypothesis testing for the assumption of the stochastic frontier model.**

| Null hypothesis | Degree of freedom | LR | *Chi square*($x^2$) tabulated value | Decision |
|---|---|---|---|---|
| $H_0:\gamma = 0$ | 1 | 66.52 | 3.8 | Reject Ho |
| $H_0:\beta_6,\beta_7 = \cdots = \beta_{20} = 0$ | 15 | 19.6 | 24.99 | Accept Ho |
| $H_0:\alpha_0,\alpha_1 = \cdots = \alpha_{13} = 0$ | 13 | 25.8 | 22.36 | Reject Ho |

Source: Own computation (2021).

The likelihood test statistic is calculated as; $LR = -2(LH_{\text{Cobb Douglas}} - LH_{\text{Translog}}) = -2[-214.3 - (-204.5)] = 19.6$ with null hypothesis stated that $H_0$: Cobb Douglas production function is appropriate and the alternative hypothesis $H_1$:Trans−log production function is appropriate, against *Chi−square*($x^2$) tabulated value at 15 degrees of freedom and 5% significance level, which was 24.99 (Table 4). Based on such a result, the null hypothesis was not rejected and concluded that the Cobb-Douglas production function was wisely fitted to the tomato data.

Fourth, does farm-level technical inefficiencies are affected or not by the farm, institutional and farmer-specific variables (determinants), that are check the overall (joint) significance of the technical inefficiency model. This was also performed by log-likelihood ratio test, $LR = -2(LH_0 - LH_1)$ where $LH_0$:- Log-Likelihood function under the stochastic production function model without explanatory variables of inefficiency effects, $LH_1$:- Log-Likelihood function under the stochastic production function model with explanatory variables of inefficiency effects. The calculated Log-Likelihood function $LR = -2(-227.2 + 214.3) = 25.8$, this was less than *Chi−square*($x^2$) tabulated value 22.36 at 13 degrees of freedom and 5% significance level (Table 4). In such a way, the null hypothesis was disfavor, and determinant variables of technical inefficiency can jointly determine variation in production of tomato output.

**Estimation of Cobb-Douglas's production function.** The stochastic frontier model estimates Cobb-Douglas tomato production functional based on five basic input variables: oxen, labor, land, seed, and tractor with the help of maximum likelihood estimation techniques. As shown in Table 5, land and labor were highly elastic, significant, and positive determinants of tomato production, i.e. a percentage increase in land and labor will result in an increase in the tomato output level of 86% and 56%, respectively, ceteris paribus (all other factors are held unchanged), and respectively. This finding consistent with the finding of [37,53–56], in a way, seed and oxen were positive and elastic implications on tomato production in the study area next to land and labor. These results were also consistent with the justification by Khan and Shoukat [57]; Bisrat [38] and Willy [58].

Under Table 5 the value of Sigma-squared ($\sigma^2$) = 0.57 and significantly different from zero at 1% level of significance, which hypothesized that perfect goodness of fit of data with the Cobb- Douglas stochastic frontier model and also the assumption of the composite error term was properly specified [55,59,60]. The estimated result of the output (Table 5) for Gamma ($\gamma$) = 0.89, which infers that 89% variation of tomato output from frontier(potential) output was due to technical inefficiency, while the remaining 11% of tomato output deviation from the potential level was because of random noises like unexpected rainfall, frost and other natural disasters beyond the control of tomato producers.

**Determinants of technical efficiency.** After estimated technical inefficiency variables by using the single-stage estimation approach of the stochastic frontier model, identified the following significant determinant factors of technical efficiency of tomato producers.

*Education.* It is measured in years of school of the household head, In this study education was a positive and statistically significant (at 1% significance level) effect on technical efficiency (Table 5). This might be due to, educated farmers were eager to disseminate technology that

**Table 5. OLS and maximum likelihood estimate for Cobb–Douglas production function for tomato.**

| Variables | Parameters | Ordinary least square estimates | | Maximum likelihood estimates | |
|---|---|---|---|---|---|
| | | Coefficients | SE | Coefficients | SE |
| Constant | $\beta_0$ | 8.23*** | 1.470 | 9.14*** | 0.501 |
| lnOxen | $\beta_1$ | -0.58** | 0.283 | 0.03** | 0.004 |
| lnLabor | $\beta_2$ | 0.42 | 0.359 | 0.56*** | 0.064 |
| lnLand | $\beta_3$ | 0.09*** | 0.012 | 0.86*** | 0.076 |
| lnSeed | $\beta_4$ | -0.89*** | 0.068 | 0.23*** | 0.041 |
| lnTractor | $\beta_5$ | 0.75 | 0.773 | 0.07 | 0.069 |
| Ln efficiency model output | | | | | |
| Constant | $\alpha_0$ | | | 4.26*** | 0.395 |
| Edu | $\alpha_1$ | | | 0.52*** | 0.037 |
| HHsize | $\alpha_2$ | | | -0.8*** | 0.085 |
| Extc | $\alpha_3$ | | | 0.32** | 0.140 |
| Offincome | $\alpha_4$ | | | -0.05*** | 0.004 |
| Trning | $\alpha_5$ | | | 0.45** | 0.205 |
| Credit | $\alpha_6$ | | | 0.92*** | 0.144 |
| Ownlvs | $\alpha_7$ | | | -0.35*** | 0.293 |
| DSTmkt | $\alpha_8$ | | | -0.01*** | 0.293 |
| Memcoop | $\alpha_9$ | | | 0.04 | 0.037 |
| DSTplot | $\alpha_{10}$ | | | 0.25 | 0.321 |
| Pest | $\alpha_{11}$ | | | -0.03*** | 0.006 |
| Expr | $\alpha_{12}$ | | | 0.24 | 0.276 |
| Sex | $\alpha_{13}$ | | | 0.36 | 0.293 |
| Variance Parameters | | | | | |
| Sigma-squared ($\sigma^2$) | | | | 0.57*** | 0.042 |
| Gamma ($\gamma$) | | | | 0.89*** | 0.049 |
| Log likelihood function | | -247.56 | | -214.3 | |
| Total sample size | | 267 | | 267 | |
| Return to scale | | 2.73 | | 2.1 | |

Source: Own computation from the survey (2021).

N.B *, ** and *** indicates level of significance at 10%, 5% and 1% respectively.

is:- they were able to receive, interpret, and disseminate new information and improved technologies such as improved seed, fertilizer, and pesticides [49]. Education also improved the unobserved labor quality and management capability of farmers, this, in turn, increase technical efficiency [38]. This finding was consistent with Woldegiorgis et al [61]; Abate et al. [60]; Bisrat [38] and Kifle [59], even though, this result is against the study by Temesgen & Ayalneh [36].

*Household size.* It is specified that the total number of people lived in one roof of the house and had a significant(at 1% significance level) and adverse effect on the technical efficiency of tomato producers (Table 5). As the number of workforce increase, will have multiple effects on producing of output however, if it was not properly managed (mismanagement) will have a hostile effect on production and productivity (diminishing marginal return of labor). This result was supported by the finding of Abebe et al [34]; Temesgen and Ayalnesh [36]; Wudineh [62], however, which was protested by the investigation of Abebe et al [34].

*Frequency of extension contact.* It had a statistically significant and positive relationship with the technical efficiency of tomato production at a 5% level of significance (Table 5).

Farmers who had to get reputed extension visits/teachings are likely to have a chance for gathering new information, techniques of production, understanding new practices, and eager to use modern inputs, which in turn will improve their technical efficiency. The result of this study is similar to the findings of Abate et al [60]; Nanii et al [63]; Daniel [64]; Dassa et al [65]. However, this finding was in contrast with the result of Abebe et al [34].

*Off-farm income*. It is a dummy variable, 1 if the household was involved in off/non-farm activities and 0 otherwise. Off-farm income has a negative and significant effect on tomato producer efficiency at a 5% level of significance in this study (Table 5). If the farmer participated in off-farm activities, they might have dedicated their time and labor to activities other than tomato production. This caused depleted the portion of time and labor force for managed and produced tomato production and finally, this might be hurt the technical efficiency of tomato producers. This result was in line with the results of Lagiso et al [33]; Hailemariam [35]; Kitila and Alemu [12] and it is in contrast with the study by Hailsellasie [66].

*Training*. It represents a dummy variable that is 1 if the farmers get training on tomato production, 0 otherwise. Training has a positive and significant effect (at 5% level of significance) on the technical efficiency of farms (Table 5). Training is an important tool for strengthening managerial capacity and improving farmers' skills in production practices from planting to harvesting and marketing [28]. Such finding was agreed with the study of Degefa [59]; Abebe et al. [34]; Haji [20]; Beyan et al. [67], even though, such a finding was different with the result of Gebregziabher [68].

*Access to credit*. It is a dummy variable indicating 1 if a farmer gets credit, 0 otherwise. Table 5 revealed that access to credit has a favorable effect on the efficiency of tomato producers at 1% level of significance. Cash requirements for purchasing inputs on time (improved tomato seed, pesticides, additional labor force, and fertilizer) and a solution to the liquidity trap resulted in the farmer being more efficient than the counterpart. The study by Daniel [69]; Bisrat [38]; Daniel [64] and Khan and Saeed [70] found that there was a positive and significant effect of credit on technical efficiency.

*Ownership of livestock*. Represent the total amount of livestock measured by Tropical Livestock Unit and it has significance at 1% level of significance (Table 5). However, ownership of livestock had a negative effect on the technical efficiency of tomato producers. This could be because, in the study area, the majority of society was agro-pastoralist, and if they owned more livestock, they would shift their practice to animal husbandry rather than vegetable production, this may be caused a reduction of the efficiency of vegetables (tomato) production. It is supported by Abebe et al. [34]; Temesgen and Ayalneg [36], however, it was in contrast with the result of Jote et al. [71].

*Distance to the market*. Represent the distance from farmer home to the nearest market in kilometers. It had a statistically significant and negative relationship with the technical efficiency of tomato production at a 1% level of significance (Table 5). If the market was a long distance from the farmers' homes, they would be unable to obtain the most recent market information (price, demand, and supply of tomatoes) and improve seed on time. As a result, the farmer who is closest to the market is technically more efficient than those who are not. The results concur with the findings of Degineh et al [33]; Wassihun et al [72]; Gebregziabher et al [68] and Asgedom et al [3].

*Pesticides*. It is dummy variable, 1 if the farmers used pesticide, 0 otherwise; it had a statistically significant and negative relationship with technical efficiency of tomato production at 1% level of significance (Table 5). Pesticide use to reduce the incidence of disease and pest infestation of tomato, notwithstanding if the farmers do not use recommended litter of pesticides per hectare of the tomato plant and they have not used safety materials during spray pesticides will lead to harm the plant and reduced technical efficiency of tomato producers and also their

**Table 6. Tomato yield gap and mean technical efficiency.**

| Variables | Observation | Mean | Standard Deviation | Minimum | Maximum |
|---|---|---|---|---|---|
| Actual Output(Kg/ha) | 267 | 5,987.23 | 5,236.30 | 4,626 | 8,753.60 |
| Technical efficiency | 267 | 0.809 | 0.167 | 0.325 | 0.999 |
| Frontier output(Kg/ha) | 267 | 7254.75 | 6823.1 | 5984.4 | 9125.3 |
| Output gap(Kg/ha) | 267 | 1,267.52 | 786.25 | 526 | 1632 |

Source: Own computation from the survey (2021).

health. This finding has parallelled the result of Aman et al. [65]and contrast with Leake et al [61], and Gebresilassie [24].

**Elasticity of production.**   Investigated return to scale is the best and wise mechanism to know the elasticity of production of tomatoes. Conceptually, returns to scale is the sum of coefficients of the major five inputs of the Cobb-Douglass production function in tomato production. Hence in this study, the estimated Cobb–Douglas production function for tomatoes was 2.1 (see Table 5), this was the sum of all inputs coefficients. This result told that there was an increasing return to scale because it was greater than a unit. It can be interpreted as when tomato producers increased all five inputs instantaneously in the Cobb Douglass production function by a unit; tomato output would increase by 2.1.

**Tomato yield gap and mean technical efficiency.**   Yield gap can be defined as an amount at which the difference between the producers produced at the frontier output and the actual output (with technical inefficiency).

The model output revealed (Table 6) that the mean technical efficiency of tomato producers in the Asaita district was 80.9%. This infers that in the short run there is room for reducing tomato production inputs by 19.1% without reducing the existing actual output of tomatoes in the study area. Alternatively, technical efficiency (19.1%) could be interpreted as, if the sampled households of tomato producers operated at full efficiency level they would increase their tomato output by 19.1% without an additional level of inputs (using the existing inputs and level of technology).

Under Table 6, both the actual and potential tomato output kilogram per hectare during the production year was 5987.23 and 7254.75, respectively with 1267.53 kilograms per hectare yield gap. Such yield gap(1267.53 kg/ha) hypothesized that tomato producers could able increase tomato output by 1267.53 kg/ha within the existing amount of inputs and technology, on average.

## Conclusions and recommendations

Numerous studies were conducted on the technical efficiency of vegetables in general and tomatoes in particular in the least developed countries including Ethiopia. Even though most of them were used a two-stage stochastic frontier model to estimate and analysis the technical efficiency of tomatoes and none of the studies were done on the technical efficiency of tomatoes in this study area. Therefore, this study was estimated the mean technical efficiency and analyzed the factors affecting the technical efficiency of tomato producers in Asaita district by using the single-stage stochastic frontier model. To achieve such objectives, econometric methods (single-stage stochastic frontier model) were used. The stochastic frontiers model revealed that the estimated values of mean technical efficiency were 80.9%, which indicated that there was an opportunity that increased tomato output level by 19.1% with the existing level of inputs and technology. The maximum likelihood estimates of the stochastic frontier for production function indicated that land, labor, tomato seed, and oxen significantly affect tomato

output. The maximum likelihood estimates of the stochastic frontier for the efficiency model indicated that education, extension contact, training, and access to credit had a positive and significant effect on technical efficiency, whereas household size, off-farm income, livestock ownership, distance to market, and pesticides had a worthy and significant effect on tomato producer technical efficiency in a study area. Based on the findings, this study recommended to responsible bodies the following issues:-the government should place more emphasis on education and educational infrastructures, provide timely and frequent training for those farmers, provide short and long-term credit opportunities; extension agents should participate in providing information and disseminating training for those farmers on land preparation, production, harvesting, and marketing, as well as providing advice on the recommended amount of pesticide per hectare and use of safety equipment when spraying pesticides; farmers should prioritize mixed farming methods over animal husbandry; the government and any concerned bodies should establish market center nearest to farmers resident. In such a way, the scope of this study was limited to the technical efficiency of tomato producers, and it could not generalized and inference the technical efficiency of vegetable producers of the nation as a whole and particularly in Afar regional state, therefore, future research should be done on technical efficiency of vegetable producers in Ethiopia as well as in Afar regional state by using this study as a reference point.

## Supporting information

**S1 Dataset. Minimal anonymized data set.**
(DTA)

## Acknowledgments

Thanks to all economics department academic staff, and research and community service vice-president of Samara University.

## Author Contributions

**Conceptualization:** Dagmawe Menelek Asfaw.

**Data curation:** Dagmawe Menelek Asfaw.

**Formal analysis:** Dagmawe Menelek Asfaw.

**Funding acquisition:** Dagmawe Menelek Asfaw.

**Investigation:** Dagmawe Menelek Asfaw.

**Methodology:** Dagmawe Menelek Asfaw.

**Software:** Dagmawe Menelek Asfaw.

**Supervision:** Dagmawe Menelek Asfaw.

**Validation:** Dagmawe Menelek Asfaw.

**Visualization:** Dagmawe Menelek Asfaw.

**Writing – original draft:** Dagmawe Menelek Asfaw.

**Writing – review & editing:** Dagmawe Menelek Asfaw.

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
