## [Decision Letter · Decision Letter 0]

5 Jul 2021

PONE-D-21-15133

Analysis of technical efficiency of small holder tomato producers in Asaita district, Afar National Regional State, Ethiopia.

PLOS ONE

Dear Dr. Dagmawe Menelek Asfaw,

Thank you for submitting your manuscript to PLOS ONE. After careful consideration, we feel that it has merit but does not fully meet PLOS ONE’s publication criteria as it currently stands. Therefore, we invite you to submit a revised version of the manuscript that addresses the points raised during the review process.

We look forward to receiving your revised manuscript.

Kind regards,

László Vasa, PhD

Academic Editor

PLOS ONE

Journal Requirements:

5. Please amend either the title on the online submission form (via Edit Submission) or the title in the manuscript so that they are identical.

7. We note that Figure 1 in your submission contain map images which may be copyrighted. All PLOS content is published under the Creative Commons Attribution License (CC BY 4.0), which means that the manuscript, images, and Supporting Information files will be freely available online, and any third party is permitted to access, download, copy, distribute, and use these materials in any way, even commercially, with proper attribution. For these reasons, we cannot publish previously copyrighted maps or satellite images created using proprietary data, such as Google software (Google Maps, Street View, and Earth). For more information, see our copyright guidelines: http://journals.plos.org/plosone/s/licenses-and-copyright.

7.1.    You may seek permission from the original copyright holder of Figure 1 to publish the content specifically under the CC BY 4.0 license. 

7.2.    If you are unable to obtain permission from the original copyright holder to publish these figures under the CC BY 4.0 license or if the copyright holder’s requirements are incompatible with the CC BY 4.0 license, please either i) remove the figure or ii) supply a replacement figure that complies with the CC BY 4.0 license. Please check copyright information on all replacement figures and update the figure caption with source information. If applicable, please specify in the figure caption text when a figure is similar but not identical to the original image and is therefore for illustrative purposes only.

8. We note you have included a table to which you do not refer in the text of your manuscript. Please ensure that you refer to Table 1 in your text; if accepted, production will need this reference to link the reader to the Table.

Reviewers' comments:

Reviewer's Responses to Questions

**Comments to the Author**

1. Is the manuscript technically sound, and do the data support the conclusions?

Reviewer #1: Yes

Reviewer #2: Partly

2. Has the statistical analysis been performed appropriately and rigorously? 

Reviewer #1: Yes

Reviewer #2: Yes

3. Have the authors made all data underlying the findings in their manuscript fully available?

Reviewer #1: Yes

Reviewer #2: Yes

4. Is the manuscript presented in an intelligible fashion and written in standard English?

Reviewer #1: Yes

Reviewer #2: Yes

5. Review Comments to the Author

Reviewer #1: The topic of the paper is very special, because it introduces tomatoes production in an African country. The author of the paper used proper literature sources which were related the topic. The methodological part of the paper is pretty good and fit to the analysed research topic. Author prepared good tables and figures which helped to understand his work. I appreciated his results and agree them. I suggest to add more figures to expalin your results.

Reviewer #2: The paper is generally well written on a topic of international interest, however, the it has some parts to be developed:

1. The paper lacks a proper motivation section. In light of the existing literature, why it is so interesting for someone to read this paper and what the paper actually adds to the current literature?

2. A literature review on technical efficiency is completely missing. How the paper is positioned in light of the literature?

3. How the results are in line with existing papers on the topic? Do they support or contradict previous results?

4. What about policy recommendations and future research avenues?

5. What about limitations of the study?

6. PLOS authors have the option to publish the peer review history of their article (what does this mean?). If published, this will include your full peer review and any attached files.

Reviewer #1: No

Reviewer #2: No

---

## [Author Response · Author response to Decision Letter 0]

20 Jul 2021

Dear Editors, 

I have special gratitude to you, for devoting your valuable time and energy to review my work entitled “Analysis of technical efficiency of smallholder tomato producers in Asaita district, Afar National Regional State, Ethiopia (PONE-D-21-15133)” and giving constructive comments and invaluable guidance. In a line with this, the author had exhaustively demonstrated and addressed questions and comments raised by editors using point-by-point responses as stated below. 

Point-by-point responses for the questions and suggestions raised by Editor

1. We note your current Data Availability Statement: “The datasets and articles used to support this study are available from the corresponding author upon reasonable request." However, in the Response to Reviewers you stated: "I have updated the data availability section in the revised manuscript and I have uploaded the minimal data set at supportive file of the revised manuscript." As you have provided a minimal data set file within your manuscript, please clarify this discrepancy between these two statements. Please do note that in the interest of long-term data availability, PLOS data policy does not allow an author to be the sole point of data access. Acceptable restrictions on public data sharing are detailed in the following link:(https://journals.plos.org/plosone/s/data-availability#loc-acceptable-data-access restrictions).

Response: Thank you dear editor, I have edited such Data Availability Statement on the online submission as “All relevant data are within the manuscript and its Supporting Information files.”

2. We also note the file “minimal anonymized data set.dta” in the manuscript in the file named “Revised Manuscript with Track Changes.docx.” Please upload “minimal anonymized data set.dta” as a Supporting Information file.

Response: Thank you dear editor, I have uploaded minimal anonymized data set.dta” as a Supporting Information file.

3. Furthermore, please also note that PLOS ONE is unable to publish previously copyrighted maps or satellite images, or images created using proprietary data. For these reasons, we cannot publish images generated by software which copyrights their output (such as Google Maps, Street View, and Earth). In order to use these images in your submission, we require explicit permission from the copyright owner to publish the figures under the CC BY 4.0 license. At this time, please kindly clarify the following regarding Figures 1: 

a) Where did the authors obtain the map in Figure 1?

b) Please state whether the map has been previously copyrighted to your knowledge.

c) If the map has been previously copyrighted, we require specific consent from the copyright holder to publish these images in PLOS ONE, under the CC BY 4.0 license. To seek permission from the copyright owner to publish your map figures under the specific Creative Commons Attribution License (CCAL), CC BY 4.0, please contact them with the following text and PLOS ONE Request for Permission form (http://journals.plos.org/plosone/s/file?id=7c09/content-permission-form.pdf):

Response: Thank you dear editor, I have remove the figure from the revised manuscript.

Thank you!

---

## [Decision Letter · Decision Letter 1]

31 Aug 2021

Analysis of Technical Efficiency of Smallholder Tomato Producers in Asaita District, Afar National Regional State, Ethiopia

PONE-D-21-15133R1

Dear Dr. Asfaw,

We’re pleased to inform you that your manuscript has been judged scientifically suitable for publication and will be formally accepted for publication once it meets all outstanding technical requirements.

Kind regards,

László Vasa, PhD

Academic Editor

PLOS ONE

Additional Editor Comments (optional):

Reviewers' comments:

Reviewer's Responses to Questions

**Comments to the Author**

1. If the authors have adequately addressed your comments raised in a previous round of review and you feel that this manuscript is now acceptable for publication, you may indicate that here to bypass the “Comments to the Author” section, enter your conflict of interest statement in the “Confidential to Editor” section, and submit your "Accept" recommendation.

Reviewer #1: All comments have been addressed

Reviewer #2: All comments have been addressed

2. Is the manuscript technically sound, and do the data support the conclusions?

Reviewer #1: Yes

Reviewer #2: Yes

3. Has the statistical analysis been performed appropriately and rigorously? 

Reviewer #1: Yes

Reviewer #2: Yes

4. Have the authors made all data underlying the findings in their manuscript fully available?

Reviewer #1: Yes

Reviewer #2: Yes

5. Is the manuscript presented in an intelligible fashion and written in standard English?

Reviewer #1: Yes

Reviewer #2: Yes

6. Review Comments to the Author

Reviewer #1: Paper have been improved by the author, who appreciated my suggestions. That is why I think this paper is ready to publish in the Journal.

Reviewer #2: The revision of the paper is well done and the responses of the author are acceptable from my side. The paper is ready for publication.

7. PLOS authors have the option to publish the peer review history of their article (what does this mean?). If published, this will include your full peer review and any attached files.

Reviewer #1: No

Reviewer #2: No

---

## [Editor Report · Acceptance letter]

14 Sep 2021

PONE-D-21-15133R1 

Analysis of Technical Efficiency of Smallholder Tomato Producers in Asaita District, Afar National Regional State, Ethiopia 

Dear Dr. Asfaw:

I'm pleased to inform you that your manuscript has been deemed suitable for publication in PLOS ONE. Congratulations! Your manuscript is now with our production department. 

Kind regards, 

on behalf of

Prof. Dr. László Vasa 

Academic Editor

PLOS ONE